# Binding of the RNA Chaperone Hfq on Target mRNAs Promotes the Small RNA RyhB-Induced Degradation in *Escherichia coli*

**DOI:** 10.3390/ncrna7040064

**Published:** 2021-09-28

**Authors:** David Lalaouna, Karine Prévost, Seongjin Park, Thierry Chénard, Marie-Pier Bouchard, Marie-Pier Caron, Carin K. Vanderpool, Jingyi Fei, Eric Massé

**Affiliations:** 1CRCHUS, RNA Group, Department of Biochemistry and Functional Genomics, Faculty of Medicine and Health Sciences, Université de Sherbrooke, 3201 Jean Mignault Street, Sherbrooke, QC J1E 4K8, Canada; d.lalaouna@ibmc-cnrs.unistra.fr (D.L.); karine.prevost@usherbrooke.ca (K.P.); thierry.chenard@usherbrooke.ca (T.C.); marie-pier.bouchard2@usherbrooke.ca (M.-P.B.); marie-pier.caron@usherbrooke.ca (M.-P.C.); 2RNA architecture and reactivity Unit, Université de Strasbourg, CNRS, ARN UPR 9002, F-67000 Strasbourg, France; 3Department of Biochemistry and Molecular Biology, Institute for Biophysical Dynamics, University of Chicago, 929 E. 57 St., Chicago, IL 60637, USA; prelist@gmail.com; 4Department of Microbiology, University of Illinois at Urbana-Champaign, 601 S Goodwin Ave., Urbana, IL 61801, USA; cvanderp@life.uiuc.edu

**Keywords:** small RNA (sRNA), Hfq, RNase E, sRNA-induced degradation, super-resolution imaging, RNA chaperone, Hfq binding site

## Abstract

Many RNA-RNA interactions depend on molecular chaperones to form and remain stable in living cells. A prime example is the RNA chaperone Hfq, which is a critical effector involved in regulatory interactions between small RNAs (sRNAs) and cognate target mRNAs in Enterobacteriaceae. While there is a great deal of in vitro biochemical evidence supporting the model that Hfq enhances rates or affinities of sRNA:mRNA interactions, there is little corroborating in vivo evidence. Here we used in vivo tools including reporter genes, co-purification assays, and super-resolution microscopy to analyze the role of Hfq in RyhB-mediated regulation, and we found that Hfq is often unnecessary for efficient RyhB:mRNA complex formation in vivo. Remarkably, our data suggest that a primary function of Hfq is to promote RyhB-induced cleavage of mRNA targets by RNase E. Moreover, our work indicates that Hfq plays a more limited role in dictating regulatory outcomes following sRNAs RybB and DsrA complex formation with specific target mRNAs. Our investigation helps evaluate the roles played by Hfq in some RNA-mediated regulation.

## 1. Introduction

Post-transcriptional regulation of gene expression by small regulatory RNAs (sRNAs) is universally found in Bacteria, Archaea, and Eukarya [1,2,3]. Bacterial sRNAs have been identified as crucial regulators that are often expressed to maintain the homeostasis of cellular pathways during environmental stress. These sRNAs are typically noncoding and smaller than 300 nucleotides. In *Escherichia coli*, ~100 sRNAs have been identified, encoded either on the chromosome or plasmids [1,4]. Usually, sRNAs act to negatively regulate target mRNAs by decreasing translation and/or increasing mRNA turnover. In some cases, sRNA base-pairing with target mRNAs activates their translation [5,6,7,8].

A common feature among sRNAs from enteric species is the requirement for the RNA chaperone Hfq for their activity [7,8,9]. This RNA binding chaperone was first discovered decades ago as a host factor essential for bacteriophage Qβ replication [10,11]. More recently, elucidation of Hfq’s role in sRNA-mediated gene regulation has been an area of intense focus. In vivo, Hfq monomers assemble to form hexamers and dodecamers, which stabilize sRNAs and modulate base-pairing with target mRNAs [12,13,14]. Several studies have shown similarities in both protein sequence and structure between bacterial Hfq and eukaryotic Sm proteins, which bind small nucleolar RNAs and are components of the spliceosome [15]. Over the years, a number of studies using Hfq co-immunoprecipitation have identified dozens of sRNAs that bind to this chaperone protein in vivo in *E. coli* and *Salmonella* [12,16,17,18]. Consistent with a role in sRNA function, Hfq has been clearly shown to enhance the stability of many sRNAs in vivo [19,20,21,22,23].

In 2002, it was first reported that Hfq facilitates pairing between an sRNA and its target mRNA [22,23]. Because of the short and imperfect pairing of sRNAs with target mRNAs, the chaperone Hfq was proposed to promote sRNA:mRNA binding through remodeling RNA structures and by increasing local concentrations of the sRNA and its target mRNA [7,8,9]. This model was supported by numerous in vitro studies [5,8,24,25,26,27,28]. Over time, these observations led to the development of a prevailing model in which the RNA chaperone Hfq is essential for sRNA:mRNA interaction [29].

Other studies have demonstrated that the role of Hfq is not restricted to mediating sRNA-dependent gene regulation. Hfq itself can serve as a negative regulator of translation initiation, competing with 30S ribosomes for accessibility to the ribosome binding site (RBS) on an mRNA [30,31,32,33]. We demonstrated that Hfq is recruited by the sRNA Spot42 to specifically bind to the translation initiation region (TIR) of *sdhC* mRNA, thereby preventing the binding of 30S ribosomal subunit [34]. A similar mechanism was shown for the sRNA SgrSregulation of the target *manX* mRNA [35]. Hfq has also been shown to promote the RNA degradosome recruitment to sRNA:mRNA complexes, inducing their rapid degradation [20,36,37,38]. This is consistent with the observation that in a Δ*hfq* strain, sRNA-mediated degradation of mRNAs is decreased or even abolished [39,40,41] Notably, we recently demonstrated that Hfq binding to the target *hdeD* is critical to promote sRNA-dependent degradation by RNase E [42]. Studies investigating global Hfq binding sites indicated additional mRNAs bound to the protein [31,32]. The presence of Hfq binding motif on target mRNAs suggests that Hfq may help recruit the sRNA:RNase E silencing complex. The chaperone was also shown to help mature the ribosomal RNAs [43]. Lastly, Hfq is also involved in RNA processing as it regulates polyadenylation-dependent mRNA decay [44,45,46]. Indeed, Hfq can stimulate the polyadenylation of mRNAs by poly(A) polymerase (PAP), which in turn triggers 3′-to-5′ degradation by an exoribonuclease [46]. In *E. coli*, this exoribonuclease can be polynucleotide phosphorylase, RNase R or RNase II [9].

With more than 25 known target mRNAs, the iron-responsive sRNA RyhB, has one of the largest regulons in *E. coli*, including *sodB* and *sdhC* transcripts [1,47,48]. In iron-rich conditions, the Ferric uptake regulator (Fur) binds Fe^2+^ and inhibits *ryhB* transcription in addition to repressing many other iron uptake genes. Upon iron starvation, Fur becomes inactive, thereby relieving the repression of RyhB and of other iron-related genes [48]. In those conditions, RyhB directly base-pairs with about 20 mRNAs encoding iron-using proteins to shut down their translation and stimulate their rapid degradation by the action of the RNA degradosome [20], a protein complex comprising the endoribonuclease E (RNase E), a 3′-5′ polynucleotide phosphorylase (PNPase), an RNA helicase (RhlB), and enolase [49]. Destabilization of mRNA targets following RyhB pairing is a consequence of both translational repression (passive degradation) and RyhB-mediated stimulation of degradation (active degradation) [50].

In this work, we used pull-down assays of MS2-tagged sRNAs in combination with high-throughput RNA sequencing to demonstrate that Hfq is not essential for certain sRNA:mRNA duplexes in vivo. Data obtained from MS2-tagged sRNAs, such as RyhB, SgrS, RybB, and DsrA, suggest they strongly bind cognate target mRNAs even in the absence of cellular Hfq. Super-resolution microscopy, with RyhB:*sodB* as an example, demonstrated that RyhB can base-pair with *sodB* mRNA in the Δ*hfq* background, with a slightly tighter affinity compared to the wild-type case. Additional data indicate that Hfq direct binding to target mRNAs is essential for sRNA-induced mRNA degradation. Mutations preventing Hfq binding on target mRNAs also prevent sRNA-induced mRNA degradation even though RyhB binds to target mRNAs and blocks their translation. We conclude that the ternary complex formed by sRNA, target mRNA and Hfq provides a scaffold for recruiting RNase E.

## 2. Results

### 2.1. Absence of Hfq Does Not Prevent MS2-Tagged RyhB Interaction with sodB and sdhC Target mRNAs In Vivo

We sought to investigate the role of the Hfq RNA chaperone in promoting sRNA:mRNA complex formation in vivo. To test this, we monitored the effect of a Δ*hfq* mutation on RyhB sRNA [40] and its ability to interact with specific target mRNAs. Thus, RyhB was MS2-tagged and expressed in vivo for 10 min, followed by MS2-RyhB pull-down. Both this MS2-RyhB construct and pull-down method were previously validated [51]. We used Northern blot assays to detect specific and well-characterized target mRNAs (*sodB* and *sdhC*). Results in Figure 1A demonstrate that MS2-RyhB can interact with endogenous cognate target mRNAs *sodB* and *sdhC* even in the absence of cellular Hfq (compare lanes two and four). Despite the reduced MS2-RyhB level in Δ*hfq* background, the co-purification of target mRNAs is greater than in WT background. The mRNA levels in the input (total RNA prior to purification) are presented in Appendix A. We noted the increased amount of *sodB* and *sdhC* input in Δ*hfq* background and explained this by reduced RyhB-mediated mRNA decay in the absence of Hfq.

We then performed a reverse pull-down assay with an MS2-tagged *sodB* construct to verify whether the sRNA RyhB would be co-purified in a Δ*hfq* background. Results shown in Figure 1B demonstrate the binding of *sodB*_130_-MS2 construct with RyhB sRNA in WT (lane two) and Δ*hfq* mutant (lane four) backgrounds. The reduced RyhB level in lane four is likely due to decreased stability of RyhB sRNA in Δ*hfq* background [20]. The mRNA levels observed before affinity purification are presented in Appendix A. These data suggest that Hfq might not be required to help MS2-RyhB pair with *sodB* or *sdhC* target mRNAs in vivo.

### 2.2. Interaction of MS2-RyhB with Some Target mRNAs Is Not Hfq-Dependent In Vivo

To further investigate the results presented above, we measured endogenous target mRNAs enrichment in a Δ*hfq* background using MS2-RyhB affinity purification coupled with RNA sequencing (MAPS) [51]. The number of reads obtained for a given mRNA target co-purified with MS2-RyhB was normalized by the FPKM method and compared to reads obtained with untagged RyhB (control). Results presented in Figure 1C,D indicate that many target RNAs (e.g., *sodB*, 3′ETS*^leuZ^*, *msrB*, and *grxD*) were efficiently recovered (Fold Change > 4) by co-purification with MS2-RyhB from the Δ*hfq* background. Altogether, these results corroborate the idea that Hfq might not be essential for pairing between RyhB sRNA and some target mRNAs.

### 2.3. Formation of RyhB:sodB mRNA Complex in the Absence of Cellular Hfq

To directly visualize the sRNA:mRNA complex formation in vivo, we utilized a previously developed super-resolution imaging approach [52]. We used a chromosomally-integrated RNase E-resistant *sodB*_130_-*lacZ* fusion [50] as the target mRNA for RyhB, and induced RyhB from the endogenous locus by addition of 2,2′-dipyridyl. *sodB*_130_-*lacZ* construct and RyhB sRNA were detected by Alexa 568 and Alexa 647-labeled DNA probes, respectively, through fluorescence in situ hybridization (FISH) [53], and imaged by 3D super-resolved single-fluorophore microscopy [54]. Probes were designed to target the *lacZ* portion on the *sodB*_130_-*lacZ* construct and to target the non-base-pairing region on RyhB (Figure 2A). Consistent with previous results, *sodB*_130_-*lacZ* remained constant after induction of RyhB (Appendix A). We determined the RyhB copy number from the super-resolution images based on the previous method [52,55]. Compared to the more abundant RyhB levels observed in the wild-type strain within a few minutes after induction, RyhB levels remained moderate and relatively constant in a Δ*hfq* mutant background even after 30 min of induction (Figure 2C and Appendix A). This was expected and consistent with previous studies demonstrating that Hfq stabilizes sRNAs in vivo [19,20,23].

RyhB:*sodB*_130_-*lacZ* mRNA complex formation was measured by determining the percentage of mRNA signal co-localizing with sRNA signal (Methods, and reference [52]). RyhB and the negative control *ptsG* mRNA, which are not regulated by RyhB, were imaged together as a negative control to account for random colocalization (Figure 2B). In both wild-type and Δ*hfq* backgrounds, the colocalization percentage of RyhB:*sodB*_130_-*lacZ* mRNA is significantly higher than random RyhB and *ptsG* mRNA colocalization (Figure 2C), suggesting complex formation between RyhB and *sodB*_130_-*lacZ* mRNA in both cases. Considering that less RyhB is present in the Δ*hfq* background, our data suggested that in the absence of Hfq, the dissociation constant (*K*_D_) is ~40% of the *K*_D_ in the wild-type case. This indicates that the binding affinity between RyhB and *sodB*_130_-*lacZ* is slightly higher (~2.5 fold) in the Δ*hfq* background, consistent with the results of the pull-down experiments described above (Figure 1A).

### 2.4. Efficient RyhB-Induced Degradation of sodB Target mRNA Is Promoted by Hfq

We next asked whether RyhB could still promote the degradation of *sodB* target mRNA in the absence of Hfq. To do so, we used Δ*hfq* mutant cells harboring a *sodB*_430_*-lacZ* transcriptional fusion [50]. We showed previously that this *sodB*_430_*-lacZ* fusion reports on RyhB-dependent RNase E-mediated decay [50]. In the Δ*hfq* background, RyhB (expressed from pBAD plasmid) repression of the *sodB*_430_*-lacZ* fusion was impaired compared to WT background (Figure 3A, compare lanes 2 and 4). Moreover, Northern blots performed on the same samples confirm the remaining presence of endogenous *sodB* mRNA in Δ*hfq* mutant as compared to WT background. As expected, the level of RyhB sRNA was reduced in the Δ*hfq* mutant. To rule out the possibility that the less efficient *sodB*_430_*-lacZ* degradation was due to the lower abundance of RyhB in the Δ*hfq* mutant, endogenous RyhB was induced (250 µM 2,2′-dipyridyl at 0.5 OD_600nm_) from WT cells (lane five) for comparison with RyhB expressed from pBAD plasmid. Even though there was less RyhB in WT cells (with DIP) compared to plasmid-expressed RyhB in the Δ*hfq* mutant, the *sodB* target mRNA was completely degraded (compare lanes four and five). These results are consistent with the idea that Hfq is required for efficient RyhB-induced degradation of *sodB* target mRNA. We performed an analogous experiment with *sdhC*_576_*-lacZ* transcriptional fusion, which also requires RyhB-dependent RNase E-mediated decay for regulation. The results also demonstrated reduced RyhB-mediated degradation of *sdhC* in the absence of Hfq (Appendix A).

### 2.5. Absence of Hfq Does Not Prevent RyhB Translation Block of sodB Target mRNA

While the results presented above suggested RyhB could bind to both *sodB* and *sdhC* target mRNAs without Hfq, it is not clear whether RyhB can block their translation under those conditions. To answer this question, we measured the β-galactosidase activity of a SodB_430_-LacZ translational fusion in both WT and Δ*hfq* backgrounds in the presence or absence of RyhB. Using this reporter, we could specifically monitor the base pairing-dependent translational repression of RyhB targets. Results in Figure 3B indicate a significant reduction in the SodB_430_-LacZ in the Δ*hfq* cells in the presence of RyhB (50%). The stronger repression observed in WT cells (70%) could be due to higher RyhB level in WT versus Δ*hfq* cells (compare lanes two and four, Northern blot) or a more active degradation of the target (Figure 3A). We also performed a similar experiment with the SdhC_258_-LacZ translational fusion. For technical reasons, we used here a shorter *sdhC* fragment to construct the translational fusion (see Methods). Appendix A shows that RyhB could still repress translation of this fusion in Δ*hfq* cells (45%). These data indicate that RyhB blocks translation of both *sodB* and *sdhC* target mRNAs in the absence of cellular Hfq.

### 2.6. Hfq Binding Site on Target mRNAs Is Essential for sRNA-Induced Degradation

We sought to investigate the role of the Hfq binding site on mRNAs targeted by sRNAs. Our lab recently showed that an AU-rich motif in the 5′-untranslated region (UTR) of *hdeD* target mRNA (close to the sRNA CyaR binding site and the RBS) is recognized by Hfq [42]. The binding of Hfq to this site in *hdeD* mRNA is essential to induce CyaR-dependent mRNA degradation. Previous work suggested that Hfq binds to a specific site within the 5′UTR of *sodB* mRNA to promote RyhB binding [19]. Moreover, Hfq binds a region in the 5′ UTR of *sdhC* mRNA, again close to the RBS [34].

To test the role of Hfq binding to target mRNAs for RyhB-dependent regulation, we mutated the previously characterized Hfq binding site on *sodB* target mRNA [19], replacing the AU-rich region upstream of the RBS with “GGCCGGC” (Figure 4A, *sodB*MH). This mutation reduced Hfq binding affinity to *sodB*_130_MH in vitro to 314 nM compared to 156 nM for wild-type *sodB* mRNA as measured by electrophoretic mobility shift assay (EMSA) (Appendix A). Moreover, in vitro probing assays on the WT *sodB*_130_ fragment indicated a clear Hfq footprint, which disappeared when using *sodB*_130_MH and Hfq (1 µM) (Appendix A). Moreover, our results suggest that secondary structures in the translation start region of *sodB*MH are not significantly altered by the mutation (Appendix A). Next, we asked whether the Hfq binding site on *sodB* and *sdhC* mRNA targets impacted their regulation by RyhB sRNA. Figure 4B shows the wild-type transcriptional fusion, *sodB*_430_-*lacZ*, was rapidly repressed after induction of RyhB. In contrast, the *sodB*_430_MH-*lacZ* construct, carrying a mutated Hfq binding site, was substantially resistant to RyhB-induced degradation (Figure 4B). We observed the same pattern when comparing the *sdhC* fusion construct (Appendix A). While the wild-type *sdhC*_576_*-lacZ* fusion was susceptible to degradation (Appendix A), the Hfq binding site mutant *sdhC*_576_MH-*lacZ* fusion (Appendix A) resisted RyhB expression (Appendix A).

These observations prompted us to monitor the kinetics of RyhB-induced mRNA degradation. Time-courses from Northern blots in Figure 4D show that the *sodB*_430_MH-*lacZ* transcriptional fusion resisted RyhB-induced degradation as compared to *sodB*_430_-*lacZ*. Endogenous *sodB* mRNA remained similarly sensitive to RyhB expression. These data suggest that the Hfq binding site located upstream of the RBS is critical for RyhB-induced degradation of both *sodB* and *sdhC* target mRNAs.

### 2.7. Hfq Binding to sodB and sdhC Target mRNAs Is Not Required for RyhB-Induced Translation Block

The results described above indicate diminished RyhB-induced degradation of both *sodB*_430_MH-*lacZ* and *sdhC*_576_MH-*lacZ* constructs as compared to wild-type controls. One possibility is that mutations in the Hfq binding sites on *sodB*_430_MH*-lacZ* and *sdhC*_576_MH*-lacZ* interfered with RyhB binding. If this were true, then mutating Hfq binding sites on *sodB* and *sdhC* would also inhibit RyhB-mediated translational regulation. To test this assumption, we measured the effect of RyhB on the Beta-galactosidase activity of SodB_430_MH-LacZ and SdhC_258_MH-LacZ constructs. Results shown in Figure 4C demonstrate that RyhB represses the translation of SodB_430_-LacZ (75%) and SodB_430_MH-LacZ (65%). We performed a similar experiment on a translational SdhC_258_MH-LacZ (Appendix A) construct with similar results (60% repression). Again, RyhB repression remained significant on SdhC_258_MH-LacZ despite the mutated Hfq binding site on the *sdhC* 5′UTR. Thus, even though the mutated Hfq binding site (MH) prevents rapid RyhB-induced mRNA degradation of *sodB* and *sdhC*, it did not interfere with RyhB-induced inhibition of translation initiation. Importantly, to demonstrate that these results were not dependent on the overproduction of RyhB from a heterologous (pBAD) promoter, we performed a similar experiment but with RyhB expressed from the endogenous locus during iron starvation induced by the iron chelator 2,2′-dipyridyl (DIP; Appendix A). In agreement with the results shown above, endogenous levels of RyhB produced during iron starvation efficiently repressed both SodB_430_MH-LacZ and SdhC_258_MH-LacZ translation. These results suggest that abrogating Hfq binding to these target mRNAs does not interfere with RyhB base pairing.

### 2.8. Hfq Must Bind sodB Target mRNA for Optimal RyhB-Promoted Degradation

The mutation Y25D in Hfq allows binding of RyhB but not of its mRNA targets [56]. Our results above prompted us to question whether Hfq Y25D could promote RyhB-induced *sodB* degradation. We determined the level of *sodB*_430_-*lacZ* activity and *sodB* target mRNA in *hfq*Y25D background in the presence and absence of *ryhB* gene. As shown in Figure 5A, the mutated *hfq*Y25D allele correlates with minimal *sodB*_430_-*lacZ* transcriptional repression by RyhB. We also observed some remaining endogenous *sodB* RNA using Northern blot analysis (compare lanes one and three), suggesting a reduced RyhB-induced *sodB* degradation. We noted that RyhB RNA level increased in Y25D background, as shown previously [56].

Moreover, we monitored the effect of RyhB on SodB_430_-LacZ translation fusion in the context of the mutated *hfq*Y25D background. Results in Figure 5B suggest that RyhB expression can significantly block target *sodB* translation by 50% in *hfq*Y25D background. We also observed the presence of a robust *sodB* mRNA band compared to WT (compare lanes one and three), which probably accounts for the higher SodB_430_-LacZ activity compared to WT. This is in agreement with previous results indicating that, in the absence of Hfq, RyhB can still bind *sodB* target mRNA and block its translation without promoting rapid degradation.

We next investigated in vivo RyhB stability in the presence of Hfq Y25D mutant. Because RyhB was previously demonstrated to co-degrade with target mRNAs [20], we reasoned that the absence of Hfq binding target mRNAs (*hfq*Y25D background) might reduce the degradation rate of RyhB. As shown in Figure 5C, the turnover of RyhB sRNA decreases significantly in the presence of Hfq Y25D as compared to the WT background. This indicates that reducing degradation of target mRNAs in a *hfq*Y25D background decreases the turnover of RyhB.

### 2.9. Effect of Hfq Deletion on the Targetome of RybB and DsrA sRNAs In Vivo

To test the role of Hfq on additional sRNA:target pairs, we monitored the effect of a null *hfq* mutant on RybB and DsrA sRNAs and their ability to interact with specific target mRNAs. These sRNAs were MS2-tagged and expressed in vivo for 10 min followed by MS2-sRNA pull-down as previously performed [51,57]. We then used Northern blot analysis to detect specific target mRNAs that co-purify with the MS2-tagged sRNA. Results demonstrate that MS2-RybB and MS2-DsrA constructs co-purify with endogenous cognate targets even in the absence of cellular Hfq (Figure 6A,D, compare lanes 2 and 4). The pattern of enrichment was comparable to MS2-RyhB (Figure 1A). The mRNA levels in the input are presented in Appendix A.

This approach allowed us to investigate the interactions of both MS2-tagged RybB and DsrA constructs with other known targets by using RNAseq. Again, we proceeded to pull-down MS2-RybB and MS2-DsrA followed by RNAseq profiling of co-purified RNAs. Results obtained with MS2-RybB (Figure 6B) and MS2-DsrA (Figure 6E) suggest efficient co-purification of target mRNAs in the *hfq* mutant background. More specifically, previously known targets of RybB (Figure 6C) and DsrA sRNAs (Figure 6F) are described. Overall, these results suggest that absence of cellular Hfq does not prevent MS2-tagged RybB and DsrA sRNAs from binding with many cognate target mRNAs in vivo. These data are also concordant with our previous results with RyhB.

### 2.10. Effect of Hfq Deletion on rpoS Target mRNA Stabilization by DsrA

Finally, we performed a time-course assay showing the effect of DsrA on the expression of the target mRNA *rpoS* in a Δ*hfq* background. DsrA was shown to increase *rpoS* mRNA stability by binding to *rpoS* 5ʹUTR and allowing translation to start [58]. We induced DsrA sRNA from a pBAD plasmid and monitored the level of the positive target mRNA *rpoS*. As shown in Figure 7, the *rpoS* mRNA level increases at a similar rate whether Hfq is present (WT) or not (Δ*hfq*). This similar kinetic of *rpoS* increase supports the idea that the absence of Hfq did not alter the in vivo binding of DsrA sRNA on target mRNA *rpoS*.

## 3. Discussion

### 3.1. Hfq Is Not Essential for RyhB, RybB, and DsrA sRNAs Complex Formation with Certain Target mRNAs

Multiple roles have been attributed to the Hfq chaperone protein. Particularly, Hfq is depicted as a key element facilitating the short and imperfect pairing between sRNAs and their cognate target mRNAs [7,8,9]. However, this model was developed primarily based on results from in vitro experiments. Contrary to this, we provide evidence from multiple approaches that cellular Hfq is probably dispensable for RyhB, RybB, and DsrA sRNAs binding to many of their target mRNAs in vivo. Using RyhB and its targets *sodB* and *sdhC* mRNAs as examples, we showed that the absence of cellular Hfq did not impair rapid (≤10 min) sRNA:mRNA complex formation (Figure 1 and Figure 2) or sRNA-mediated translational regulation of targets (Figure 3). Instead, loss of Hfq binding on target mRNAs (Figure 4D and Figure 5A) reduced RyhB-dependent mRNA degradation. Looking more broadly at other well-characterized sRNAs, i.e., RybB and DsrA, we showed that Hfq was not required for efficient interaction with some target mRNAs in vivo (Figure 6 and Appendix A). Moreover, Figure 7 shows that DsrA sRNA binds and stabilizes *rpoS* target just as rapidly in the absence or presence of Hfq. Thus, Hfq did not seem to speed up the process of DsrA:*rpoS* pairing. Finally, using the recently developed super-resolution imaging method [52], which allowed us to image and quantify sRNA:mRNA complex formation, we confirmed that endogenously-expressed RyhB (induced by iron starvation) and target mRNA *sodB* could form a complex in the presence or absence of Hfq in vivo (Figure 2). Remarkably, the RyhB:*sodB* complex has only a slightly lower *K*_D_ (higher affinity) in the Δ*hfq* background compared to wild-type, roughly consistent with the results from our MS2 pull-down assays.

Previous work had already suggested that Hfq might not be as crucial for sRNA:mRNA complex formation as the prevailing model postulates. In 2015, Fei et al. demonstrated by super-resolution microscopy that the absence of cellular Hfq only marginally decreased SgrS sRNA binding to *ptsG* mRNA and that decreased sRNA stability mostly accounted for reduced sRNA:mRNA complex formation [52]. This is consistent with Hfq being involved in the stabilization of many sRNAs in vivo [20,22,23]. Moreover, we previously showed that MS2-tagged *sdhC* mRNA could efficiently co-purify with Spot42 in a Δ*hfq* background [34]. Earlier observation also indicated that RyhB sRNA could efficiently repress translation of a SodB_430_-LacZ reporter in a Δ*hfq* background [50]. We also had similar results for the regulation of *hdeD* mRNA by CyaR in the absence of Hfq [39]. At least two other groups have demonstrated that the sRNA DsrA can activate *rpoS* independently of Hfq [59,60]. Importantly, we demonstrated in this study that endogenously-expressed RyhB can base pair with and regulate target mRNAs in the absence of Hfq or when Hfq binding sites on the target were mutated.

### 3.2. Hfq Binding Sites Are Not Required for sRNA:mRNA Complex Formation

Recently, it was proposed to classify sRNAs according to the face of the Hfq hexamer to which they bind [56]. Three distinct RNA binding faces have been characterized: a proximal face, a rim, and a distal face. RyhB, RybB, and DsrA have been categorized as class I sRNAs, binding Hfq proximal face and rim. In the case of class I sRNAs, the unoccupied distal face will interact with ARN motifs displayed by the target mRNA. Both *sodB* and *sdhC* mRNAs are regulated through Hfq binding to an AU-rich sequence located within their 5′ UTRs. Several ARN motifs are present on both *sodB* and *sdhC* mRNAs, suggesting that the Hfq distal face could efficiently recognize these target mRNAs and form RyhB:mRNA:Hfq ternary complexes. Such ternary complexes have been suggested to enable base-pairing, mRNA regulation, and degradation [56].

We used RyhB sRNA with both *sodB* and *sdhC* mRNAs as models to test the requirement for Hfq:sRNA:mRNA ternary complex formation on sRNA-mediated target regulation. We had already identified an Hfq binding site close to the translation initiation region of the *sdhC* mRNA involved in translational regulation [34]. In vitro work by another group suggested that Hfq binding to *sodB* mRNA remodeled secondary structures to facilitate RyhB:*sodB* mRNA pairing [19]. This last report was consistent with previous in vitro work demonstrating the rapid formation of sRNA:mRNA complexes in the presence of Hfq [22,23]. However, our in vivo results indicate that mutation of the Hfq binding sites (MH) on *sodB* and *sdhC* mRNAs did not significantly reduce the ability to base-pair efficiently with RyhB. Indeed, we could detect RyhB repression of translation on SodB_130_MH-LacZ and SdhC_258_MH-LacZ constructs (Figure 4C and Appendix A). Altogether, these results suggest that even though Hfq promotes RyhB sRNA:mRNA pairing in vitro, this is not an essential in vivo function for *sodB* and *sdhC* target mRNAs. Moreover, the absence of Hfq or mutation of Hfq binding site on *hdeD* target mRNA does not prevent the sRNA CyaR (Class II, [56]) binding and *hdeD* translational block [42]. SgrS, which is an intermediate sRNA between Class I and Class II [56], also binds in vivo to previously known targets such as *ptsG* in the absence of Hfq (Appendix A). Thus, our results suggesting that Hfq is not essential for RyhB, RybB, and DsrA sRNA:mRNA pairing is not only restricted to Class I sRNAs.

### 3.3. Hfq Binding to Target mRNA Is Crucial to Induce Rapid sRNA-Dependent Degradation

Remarkably, our work confirms a function for Hfq binding to target mRNAs to promote sRNA-induced mRNA degradation. Our data stressed the importance of Hfq binding to target mRNA (*sodB*_130_MH, *sdh*_576_MH, or Hfq Y25D) for sRNA-induced mRNA degradation. Even though we observed reduced degradation of the target mRNA under conditions of reduced Hfq binding, the sRNA remained able to block translation. These data strongly support the idea that RNase E activity requires Hfq binding to both the sRNA and the target mRNA for sRNA:mRNA pairing to induce target mRNA degradation. This could help stabilize a quaternary sRNA:mRNA:Hfq:RNase E complex. Similar to CLASH [61] and RIL-seq [62] data, Hör and Vogel observed that even if two different bait proteins were used (RNase E and Hfq, respectively), the number of sRNA:mRNA chimera is comparable, suggesting that both RNA binding proteins act together [63]. Perhaps Hfq binding to both RNAs in the complex promotes a stronger or longer-lasting interaction with RNase E, promoting rapid target degradation. This is reminiscent of our data on the sRNA CyaR, which induces degradation of *hdeD* target mRNA. CyaR requires an Hfq binding motif on *hdeD* to induce degradation but not to block translation [42]. The requirement for Hfq binding to the target mRNA could create a quality control step by conferring an additional level of specificity for RNase E cleavage. In addition to a base-pairing interaction, the interaction of Hfq with both RNAs in the complex could be required to activate RNase E. Such a “double-check” mechanism might prevent degradation of mRNAs that spuriously base pair with non-cognate sRNAs.

## 4. Conclusions

Despite the short and imperfect base-pairing between an sRNA and its target mRNAs, Hfq is not always required in vivo for assisting pairing between these two molecules. Because the absence of Hfq decreases the stability of a large group of sRNAs, it was often interpreted that Hfq promotes sRNA regulation by favoring base pairing in vivo. We propose that reduced sRNA stability largely accounts for the absence of effect on cognate target mRNAs in *hfq* mutants. Although Hfq is not necessary for sRNA-mediated translational inhibition, Hfq is crucial to induce rapid RNase E-dependent mRNA decay. We postulate that Hfq enables the formation of a quaternary ribonucleoprotein complex through the direct interaction with all partners (sRNA, target mRNA, and RNase E). Further characterization of this mechanism will be key to fully understand the precise role of Hfq in promoting the degradation of target mRNAs.

## 5. Materials and Methods

### 5.1. Growth Conditions

All experiments used derivatives of *E. coli* K12 substrain MG1655 *lacX74* [40]. Strains used in this study are listed in Appendix A.

Cells were grown at 37 °C in a rich medium (Lysogeny broth, LB) with agitation (220 rpm). sRNAs or MS2-sRNAs expression was induced by adding 0.1% arabinose (strains carrying pBAD-*sRNA*, pBAD-MS2-*sRNA*, pGD3-*ryhB* or the control vectors pNM12 and pGD3). Endogenous expression of the *ryhB* gene was induced by the addition of 250 µM 2,2′-dipyridyl (DIP; wild-type strains) at indicated OD_600nm_. Ampicillin (pBAD, pFRΔ, or pRS1551 derivates) and chloramphenicol (pGD3 derivates) were used at a final concentration of 50 µg/mL and 30 µg/mL, respectively. Strains constructed by P1 transduction were selected for the appropriate antibiotic-resistant marker.

### 5.2. RNA Extraction and Northern Blot Analysis

Overnight cultures were diluted 1000-fold in 50 mL of fresh LB medium and grown at 37 °C. 0.1% arabinose or 250 µM 2,2′-dipyridyl (DIP) was added when indicated. Total RNA was extracted according to the hot phenol protocol [64]. Northern blots were performed as previously described [34,50]. In brief, 5–10 µg of total RNA was loaded on a polyacrylamide gel (5%–10% acrylamide 29:1, 8 M urea) or 20 µg on an agarose gel (1%, MOPS 1x). Then, RNA was electro-transferred to a Hybond-XL membrane (Cytiva Life Sciences^TM^ Amersham^TM^, Malborough, MA, USA) for a polyacrylamide gel or transferred by capillarity on a Biodyne B membrane (Pall corporation, Port Washington, NY, USA) for an agarose gel. Crosslinking was performed by UV (1200 J, Stratagene UV Stratalinker 1800 Crosslinker, La Jolla, CA, USA). Radiolabeled DNA probes and RNA probes used in this study are described in Appendix A. Membranes were then exposed to phosphor storage screens and analyzed using a Typhoon Trio (GE Healthcare) instrument. Quantification was performed using the Image studio lite software (LI-COR, version 5.2). Results reported here correspond to data from at least two independent experiments.

### 5.3. Proteins Extraction and Western Blot Analysis

Protein extraction was performed using the following protocol. Cold TCA solution was added to cells (5% final concentration), and the mixture was placed on ice for 10 min. After precipitation (15,000 g, 10 min), the protein precipitate was washed with 80% acetone (twice). Western blot analysis was performed as previously reported [34]. Proteins were resuspended in protein-loading gel electrophoresis buffer, followed by separation on SDS-PAGE gel and transfer to nitrocellulose membrane (Cytiva Life Sciences^TM^ Amersham^TM^ Protran^TM^ NC, Malborough, MA, USA). The monoclonal ANTI-FLAG^®^ M2 antibody produced in mouse (Millipore Sigma, Burlington, MA, USA) was used at a dilution of 1:1000. The IRDye 800CW-conjugated goat anti-rabbit secondary antibody (Li-Cor Biosciences, Lincoln, NE, USA) was used at a dilution of 1:15,000. Western blots were revealed on an Odyssey infrared imaging system (Li-Cor Biosiences, Lincoln, NE, USA), and quantification was performed using the Odyssey (Li-Cor Biosciences, version 3.0 software). The results reported represent data of at least two independent experiments.

### 5.4. β-Galactosidase Assays

Kinetics assays for β-galactosidase activity were performed as described previously using a SpectraMax 250 microtitre plate reader (Molecular Devices, Sunnyvale, CA, USA) [26]. Briefly, overnight bacterial cultures incubated at 37 °C were diluted 1000-fold in 50 mL of fresh LB medium and grown with agitation (220 rpm) at 37 °C. When required, expression of respective sRNAs was induced by the addition of 0.1% arabinose at an OD_600nm_ of 0.1 or 250 µM DIP. Specific β-galactosidase activities were calculated using the formula Vmax/OD_600nm_ when cells reached an OD_600nm_ between 0.5 and 0.8 (exponential phase of growth). Data represent the mean of three independent experiments (± standard deviation, SD). See Appendix A and Methods for details on the construction of *lacZ* fusions.

### 5.5. MS2-Affinity Purification

We performed MS2-affinity purification as described previously [39,51]. The bacterial strains were grown to an OD_600nm_ of 0.5 (100 mL), at which point 0.1% arabinose was added to induce the expression of MS2-RNA/control RNA during 10 min. Cells were chilled for 10 min on ice. At this point, RNA was extracted following the hot-phenol protocol from 600 μL of culture (input). The remaining cells were then centrifuged, resuspended in 1mL of buffer A (20 mM Tris-HCl at pH 8.0, 150 mM KCl, 1 mM MgCl2, 1 mM DTT), and centrifuged again. Cells were resuspended in 2 mL of buffer A and lysed using a French Press (430 psi, three times) (Thermo Electron corporation, Needham Heights, MA, USA). The lysate was then cleared by centrifugation (17,000× *g*, 30 min, 4 °C). At this step, 20 μL of the soluble fraction was mixed with 20 μL of protein sample buffer (input). The remaining soluble fraction was subjected to affinity chromatography (all steps performed at 4 °C). The column was prepared by adding 75 μL of amylose resin (New England Biolabs, Ipswich, MA, USA) to Bio-Spin disposable chromatography columns (Bio-Rad, Mississauga, ON, Canada). The column was washed with 3 mL of buffer A. Next, 100 pmol of MS2-MBP protein was immobilized on the amylose resin, and the column was washed with 2 mL of buffer A. The cleared lysate was then loaded onto the column, which was washed with 5 mL of buffer A. RNA and proteins (output) were eluted from the column with 1 mL of buffer A containing 15 mM maltose.

Eluted RNA was extracted with phenol-chloroform, followed by ethanol (3 vol) precipitation of the aqueous phase in the presence of 20 mg of glycogen. For protein isolation, the organic phase was subjected to acetone precipitation. RNA samples were then analyzed by Northern blot and protein samples by Western blot. The reported results correspond to data from two independent experiments.

### 5.6. MS2-Affinity Purification Coupled with RNA Sequencing

We performed MAPS using MS2-RyhB, MS2-RybB, or MS2-DsrA constructs in Δ*sRNA* Δ*hfq* backgrounds (*n* = 2). As described above, cells were harvested in the exponential phase of growth (OD_600nm_ = 0.5; 100 mL). After MS2 affinity purification, samples were treated with TURBO DNase (Ambion, ThermoFisher Scientific, Waltham, MA, USA). cDNA libraries were prepared with ScriptSeq v2 RNAseq Library Preparation Kit (Illumina, San Diego, CA, USA). Samples are then sequenced on a MiSeq Sequencing System (Illumina, San Diego, CA, USA). We used Galaxy Project [65] and UCSC Microbial GenomeBrowser [66] to analyze and visualize data (see Lalaouna et al. 2017 for more details [39]). Abundances were reported in Fragments Per Kilobase of transcript per Million mapped reads (FPKM) [67]. Analyses were performed using the R environment for statistical computing (Version 3.6.1). We verified the correlation of FPKM between MS2-tagged samples produced with the same sRNA. We calculated *p*-values for each gene with each separate sRNA using the exactTest function with no dispersion from the edgeR package [68], indicating the significance of the difference between the means of two negative binomial distributions [69]. We then corrected for multiple testing using the p.adjust function with the Benjamini–Hochberg False Discovery Rate adjustment method [70]. We averaged the FPKM of the tagged and untagged samples to calculate an average enrichment for every genomic region. Volcano plots were generated for each sRNA using the log2 of the enrichment and the −log10 of the associated q-value.

### 5.7. Super-Resolution Imaging

Two probes for fluorescence in situ hybridization (FISH) were designed to target RyhB sRNA. 5′ ends of these probes were amine-modified and were labeled with Alexa 647 dyes (Figure 2A). Similarly, 14 probes labeled with Alexa 568 dyes were targeting the *lacZ* region of the *sodB_130_*-*lacZ* mRNA or the *ptsG* mRNA in the negative controls for colocalization analysis.

The hybridization protocol was described previously [52]. Briefly, overnight cell cultures were diluted by around 1:200 and incubated in MOPS-rich media at 37 °C in a shaker. When OD_600nm_ reached about 0.3, 2.2′-dipyridyl was added to a final concentration of 250 µM. After 4 min for wild-type or 30 min for Δ*hfq* cells, cells were fixed by 4% formaldehyde for 30 min, washed, and permeabilized by 70% ethanol. 60 µL of these permeabilized cells were washed with FISH wash solution (10% formamide in 2X SSC) and resuspended in 15 µL of hybridization buffer (10% dextran sulfate, 10% formamide in 2X SSC) containing the specific probes. The next day, cells were washed three times with FISH wash solution with 30 min intervals. Then, cells were postfixed with 4% formaldehyde and put on 8-well chambered coverglass (Nunc^TM^ Lab-Tek^TM^ 155409) that was coated with 0.1% poly-L-lysine (P8920, Millipore Sigma, Burlington, MA, USA).

Details of this technique were described in previous reports [54,56]. Briefly, we used an inverted optical microscope (Nikon Ti-E with 100X NA 1.49 CFI HP TIRF oil immersion objective) with a red laser (647 nm, 120 mW, Cobolt MLD), a yellow laser (561 nm, 150 mW, Coherent Obis LS) and a violet laser (405 nm, 25 mW, CrystaLaser) fiber coupled to the microscope body. Laser lines were reflected by a dichroic mirror (Chroma zt405/488/561/647/752rpc-UF3) having near-TIRF excitation. The emission signal was collected by the objective, filtered by emission filters (Chroma ET700/75m for the red channel, Chroma ET595/50m for the yellow channel), and imaged on a 1024 × 1024 EMCCD camera (Andor iXon Ultra 888). For 3D imaging, a cylindrical lens with 10 m focal length (CVI RCX-25.4-50.8-5000.0-C-415-700) was inserted in the emission path (Huang et al. 2008). Violet laser power was modulated to keep the number of blinking-on spots above 50% of the number of cells in the field of view. When the number of blinking-on spots reaches less than this, even with the maximum violet laser power, the acquisition was terminated. The power density lasers on the sample were about 2000 W·cm^−2^ for the red laser, 4300 W·cm^−2^ for the green laser, and the maximum power density for the violet laser was about 130 W·cm^−2^. Wild-type *E. coli* cells stained for *ptsG* mRNA labeled with both Alexa 568 and Alexa 647 dyes were mixed with cells of interest, serving as markers for correcting the chromatic shift between Alexa 568 and Alexa 647 channels.

Prior to imaging, 500 µL of imaging buffer (10 mM NaCl, 50 mM Tris and 10% glucose in 4X SSC, pH = 8.0) was mixed with 30 Units of glucose oxidase (G2133-10KU), and 454.5 Units of catalase (219001), (Millipore Sigma, Burlington, MA, USA). This mixture was put on the sample and incubated for 25 min before imaging.

### 5.8. Copy Number Calculation and Colocalization Analysis

A density-based cluster analysis algorithm was used to detect clusters from the raw data [71]. Briefly, a set of points having more than defined of spots (Npt) within a sphere of radius (Eps) starts to form a cluster, and it can expand by the distance cut-off, Eps. Npt = 2 and Eps = 25 nm were empirically determined. The detailed procedures of RNA copy number calculation and colocalization analysis were previously described (Fei et al. 2015). Briefly, super-resolution images of cells containing basal levels of RyhB (without induction by 2.2′-dipyridyl) went through cluster analysis, providing the numbers of localization spots per cluster. Since there are few RNAs per cell in this case, they were assumed not to be overlapping, so these clusters were considered single RNAs. Then the negative binomial distribution of these numbers was created, giving a reference matrix by which the RNA copy number could be calculated from the number of localization spots from a given cluster, in an induced cell where multiple RNAs can well overlap to form a single cluster. Δ*ryhB* cells were stained with RyhB FISH probe to give the nonspecific binding background of FISH signal. This background was used to correct the induced RyhB data for calculating the RNA copy number per cell.

Colocalization percentage between mRNA and sRNA was obtained by calculating the percentage of mRNA clusters within a cut-off distance of sRNA clusters over mRNA clusters away from any sRNA clusters by the distance cut-off. 50 nm was chosen as the distance cut-off, considering the 50 nm axial resolution of our imaging scheme.

## Figures and Tables

**Figure 1 ncrna-07-00064-f001:**
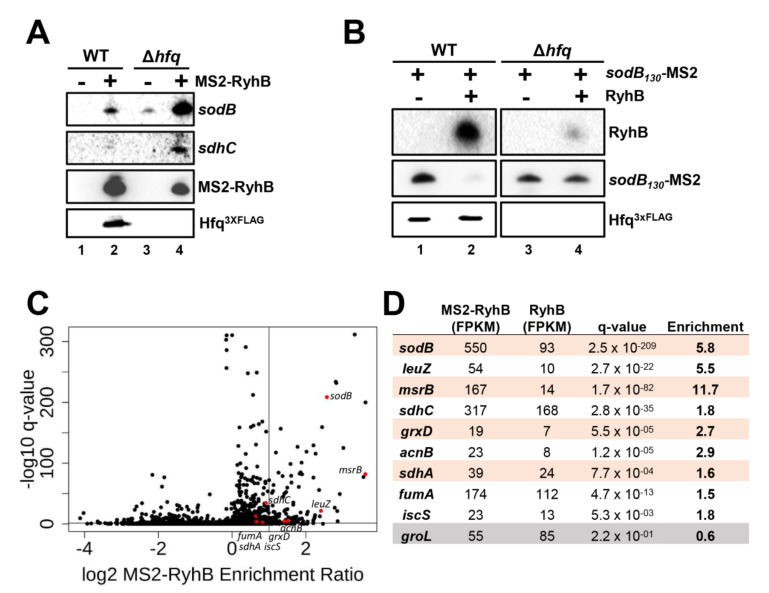
Hfq is not essential for the pairing between RyhB and its target mRNAs. (**A**) Affinity purification of the MS2-RyhB construct with previously known target mRNAs (*sodB* and *sdhC*) in WT and Δ*hfq* strains. Untagged sRNA RyhB was used as a control. The expression of both MS2-RyhB (+) and RyhB (Control; -) transcripts were induced with 0.1% arabinose for 10 min at OD_600nm_ of 0.5 (exponential phase). See also Appendix A for input. (**B**) Co-purification of RyhB with *sodB*_130_-MS2 construct in a Δ*hfq* background. The *sodB*_130_-MS2 construct was cloned into a pFR∆ plasmid, under the control of *sodB* endogenous promoter. Expression of RyhB was induced with 0.1% arabinose for 10 min at OD_600nm_ of 0.5 (pGD3-*ryhB* (+), empty vector pGD3 was used as control (−)). See also Appendix A for input. Northern blots were performed with DNA probes specific for respective RNAs, and anti-FLAG antibodies were used for Hfq^3xFLAG^ Western blot analysis. Results are representative of at least two independent experiments. (**C**) Volcano plot of the transcript enrichment ratio from MS2-RyhB/RyhB pull-down followed by RNAseq (MAPS) performed in Δ*hfq* strains (N = 3). The expression of MS2-RyhB and untagged RyhB (Control) was induced with 0.1% arabinose for 10 min at OD_600nm_ of 0.5. Black dots are all transcripts detected through RNAseq. Red dots are previously known target mRNAs of RyhB sRNA. Horizontal line is the q-value at 0.05. Vertical line represents 2x enrichment. (**D**) RNAseq was performed on RNAs co-purified with MS2-RyhB (MAPS) extracted from Δ*hfq* background cells. Reads of previously known RyhB targets (red dots in 1C) were normalized by Fragments Per Kilobase of transcript per Million mapped reads (FPKM) and compared to reads obtained with RyhB (Control). The *groL* mRNA is a non-target control.

**Figure 2 ncrna-07-00064-f002:**
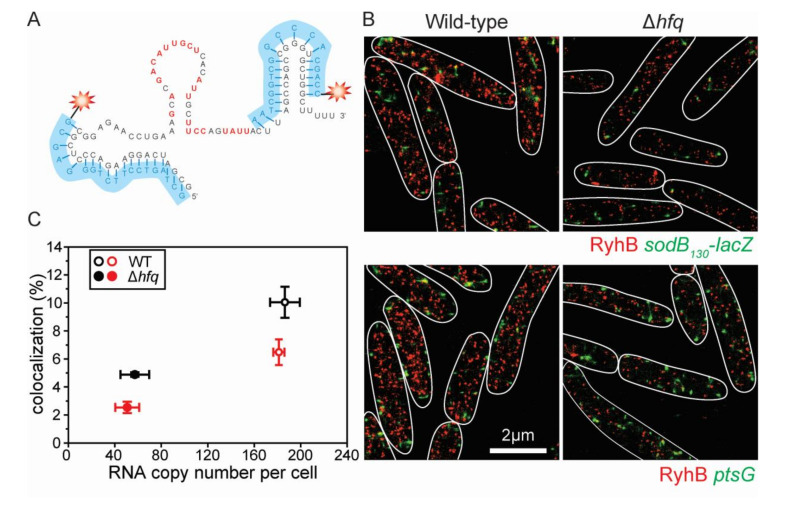
Super-resolution imaging of RyhB-*sodB* complex formation in vivo. (**A**) Two FISH probes labeled with Alexa 647 dyes were used to visualize RyhB sRNA molecules. Sequences with red letters represent characterized *sodB* binding sites on RyhB. (**B**) Four representative super-resolution images are shown here. (Upper panel) WT and Δ*hfq* cells were stained for RyhB (red, Alexa 647 dye) and *sodB_130_*-*lacZ* (green, Alexa 568 dye). (Lower panels) The same cells were stained for RyhB and *ptsG* mRNA (negative control). (**C**) The four cases are plotted for RyhB RNA copy number per cell versus the colocalization percentage between sRNA and mRNA. Red markers (lower colocalization) represent the case for RyhB and *ptsG*, while black markers (higher colocalization) for RyhB and *sodB*_130_-*lacZ*. Error bars represent the standard deviation of three biological replicates, each containing at least 200 cells. We used a total number of 787 WT cells for RyhB and *sodB_130_*, 669 WT cells for RyhB and *ptsG*, 812 Δ*hfq* cells for RyhB and *sodB*_130_, and 751 Δ*hfq* cells for RyhB and *ptsG* were imaged. See also Appendix A.

**Figure 3 ncrna-07-00064-f003:**
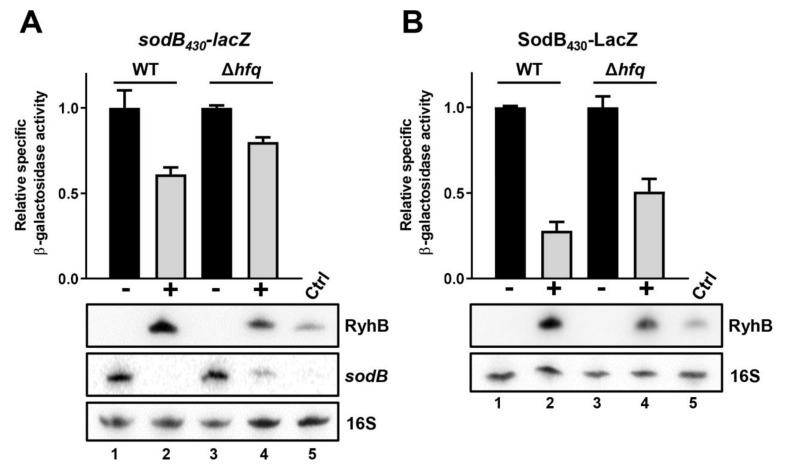
The absence of Hfq affects RyhB-mediated regulation of *sodB* mRNA. β-galactosidase activity of (**A**) *sodB*_430_-*lacZ* transcriptional and (**B**) SodB_430_-LacZ translational fusions in WT and Δ*hfq* backgrounds in presence or absence of RyhB. Strains carry either an empty vector pNM12 (black bars (−)) or a pBAD-*ryhB* (grey bars (+)). The expression of *ryhB* was induced by the addition of 0.1% arabinose when cells reached an OD_600nm_ of 0.1. Samples were taken at an OD_600nm_ of 0.5. Northern blot assays were performed at the same time to monitor the level of RyhB sRNA and *sodB* mRNA. 16S rRNA was used as a loading control. As a control (Ctrl, lane five), we monitored the endogenous expression of RyhB in WT background, which was induced by the addition of 250 µM DIP when cells reached an OD_600nm_ of 0.1. Samples were taken at an OD_600nm_ of 0.5. Data are representative of three independent experiments ± SD.

**Figure 4 ncrna-07-00064-f004:**
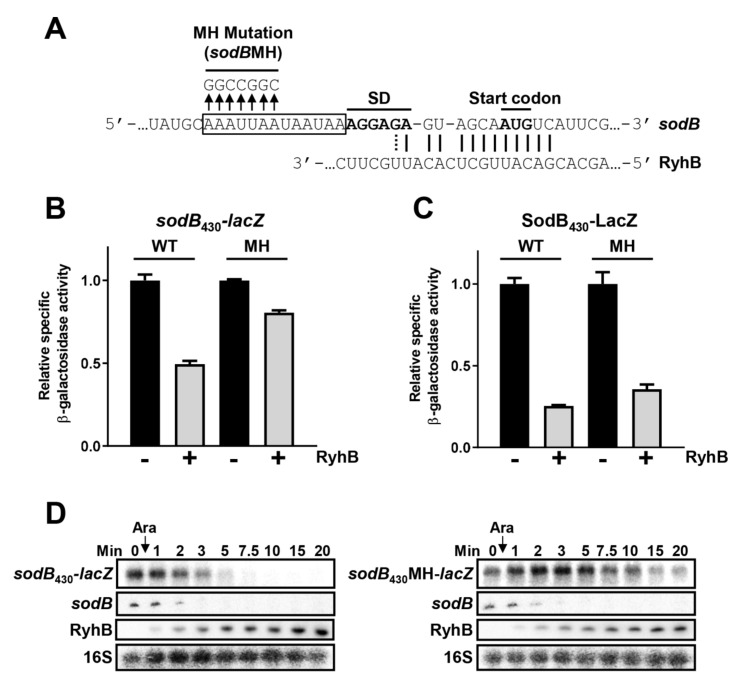
The Hfq binding site on *sodB* mRNA is required for rapid RyhB-induced mRNA decay. (**A**) Description of RyhB sRNA binding to *sodB* mRNA. SD: Shine and Dalgarno sequence. The boxed sequence is wild-type Hfq binding site. Arrows represent the mutated nucleotides present in *sodB*MH constructs. β-galactosidase activity of (**B**) *sodB*_430_-*lacZ* and *sodB*_430_MH-*lacZ* transcriptional and (**C**) β-galactosidase activity of SodB_430_-LacZ and SodB_430_MH-LacZ translational fusions in presence or absence of RyhB (pBAD-*ryhB*, grey or pNM12, black). The expression of *ryhB* was induced by the addition of 0.1% arabinose when cells reached an OD_600nm_ of 0.1. Samples were taken at an OD_600nm_ of 0.5. Data are representative of three independent experiments ± SD. (**D**) Northern blots showing *sodB*_430_*-lacZ* or *sodB*_430_MH*-lacZ* constructs level in a Δ*ryhB* background. Expression of *ryhB* gene from a pBAD promoter was induced with 0.1% arabinose when cells reached an OD_600nm_ of 0.4. At indicated time points, total RNA was extracted. A *lacZ* probe was used to specifically visualize *sodB*_430_-*lacZ* mRNA. The endogenous *sodB* mRNA was used as a positive control of RyhB-induced cleavage. 16S rRNA is shown as a loading control. Results are representative of at least two independent experiments.

**Figure 5 ncrna-07-00064-f005:**
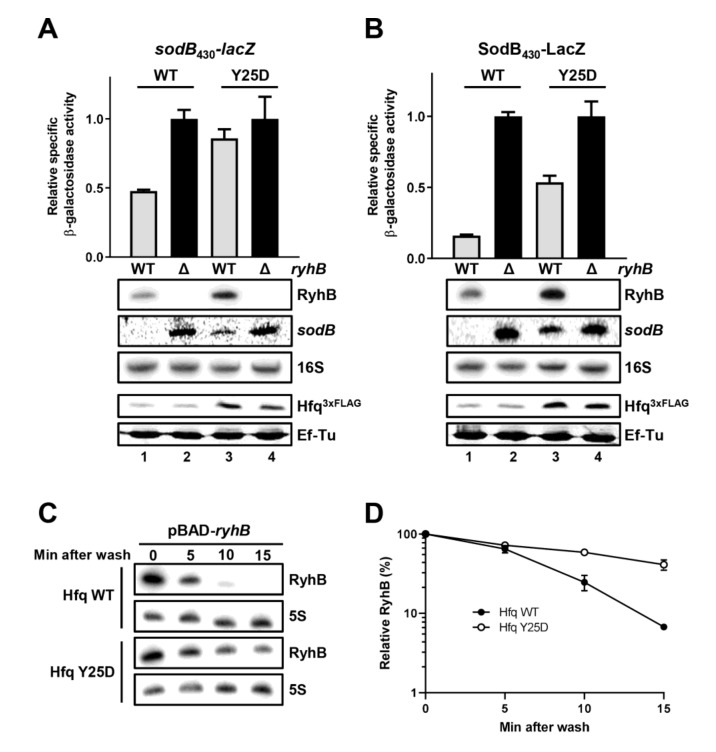
The mutated Hfq Y25D RNA chaperone prevents RyhB-induced cleavage but not translation block of *sodB* mRNA. (**A**) β-galactosidase activity of *sodB*_430_-*lacZ* transcriptional and (**B**) SodB_430_-LacZ transcriptional fusions in WT Hfq or mutated Hfq Y25D backgrounds, in presence (WT, grey) or absence (∆, black) of RyhB. The expression of *ryhB* was induced by addition of 250µM DIP when cells reached an OD_600nm_ of 0.1. Samples were taken at an OD_600nm_ of 0.5. Northern blots (RyhB and *sodB* probes) and Western blots (anti-FLAG antibodies) were performed at the same time to monitor the level of RyhB sRNA, *sodB* mRNA and the level of Hfq^3xflag^ and HfqY25D^3xflag^ protein. 16S rRNA and Ef-Tu were used as loading controls. Data are representative of three independent experiments ± SD. (**C**) RyhB synthesis was monitored in WT and Y25D mutant backgrounds. Cells were grown in LB to an OD_600nm_ of 0.5. RyhB expression from a pBAD plasmid was induced by addition of 0.1% arabinose for 15 min. The culture was then centrifuged to wash out remaining arabinose and the pellet was resuspended in an equal volume of LB with 0.2% glucose to completely suppress RyhB expression. Samples were taken at indicated intervals and total RNA was extracted. 5S rRNA was used as a loading control. Results are representative of at least two independent experiments. (**D**) Quantification of the RyhB RNA level normalized to the 5S rRNA level shown in (**C**). Half-life for RyhB sRNA in WT background is 5.3 min and Hfq Y25D is 7.9 min. Error bars represent the standard deviation of three biological replicates.

**Figure 6 ncrna-07-00064-f006:**
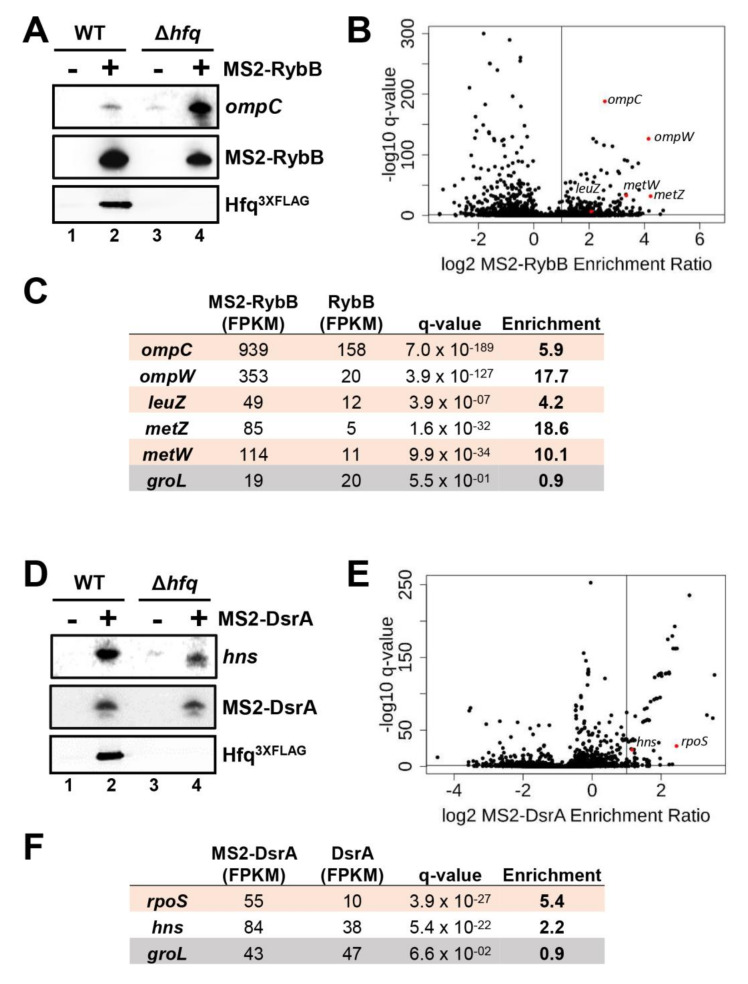
Hfq is not essential for RybB and DsrA sRNAs pairing with target mRNAs in vivo. (**A**) Visualization of the previously known target mRNA *ompC* after co-purification with MS2-RybB construct in WT and Δ*hfq* strains. Untagged sRNA RybB was used as control. The expression of both MS2-sRNA (+) and RybB (Control; -) transcripts were induced with 0.1% arabinose for 10 min at OD_600nm_ of 0.5. Northern blots were performed with DNA probes specific for respective RNAs, and anti-FLAG antibodies were used for Hfq^3xFLAG^ Western blot analysis. Results are representative of two independent experiments. See also Appendix A. (**B**) Volcano plot of the transcript enrichment ratio from MS2-RybB/RybB affinity purification coupled with RNAseq (MAPS) performed in Δ*hfq* strains (*n* = 2). The expression of MS2-RybB and untagged RybB (Control) was induced with 0.1% arabinose for 10 min at OD_600nm_ of 0.5. Black dots are all transcripts detected through RNAseq. Red dots are previously known target mRNAs of RybB sRNA. The horizontal line is the q-value at 0.05. The vertical line represents 2× enrichment. (**C**) Enrichment of previously known RybB targets was normalized by FPKM and compared to reads obtained with RybB (Control). Similar experiments have been performed with MS2-DsrA construct as bait (**D**–**F**).

**Figure 7 ncrna-07-00064-f007:**
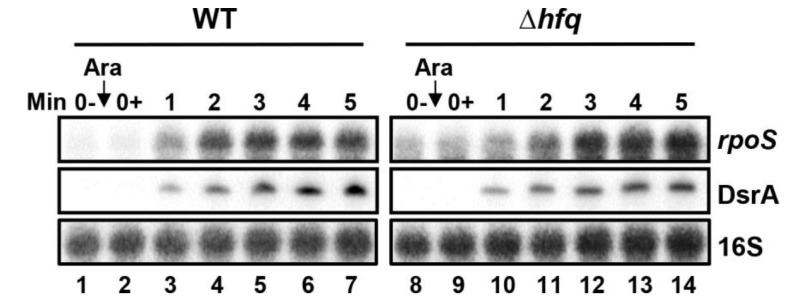
Time-course assays showing the activation of *rpoS* mRNA in the presence of DsrA in both WT and Δ*hfq* backgrounds. DsrA sRNA was induced from pBAD-*dsrA* by the addition of 0.1% arabinose at OD_600nm_ of 0.4. Northern blot assays were performed at different time points to monitor the level of DsrA sRNA and *rpoS* mRNA. 16S rRNA was used as a loading control. Results are representative of at least two independent experiments.

## Data Availability

*GEO* accession number of MAPS data (MS2-RyhB, MS2-DsrA, and MS2-RybB) is GSE113593.

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
