# Peer review of "Binding of the RNA Chaperone Hfq on Target mRNAs Promotes the Small RNA RyhB-Induced Degradation in Escherichia coli"

_ncrna, 2021, doi:10.3390/ncrna7040064_

Round 1

Reviewer 1 Report

Most base-pairing sRNAs, along with an RNA chaperon Hfq, regulate translation and/or stability of target mRNAs in enterica bacteria. It is well established that the primary role of Hfq is to facilitate sRNA-mRNA base pairing at least in vitro. It is also well known that Hfq is essential for the RNase E-dependent degradation of sRNA-mRNA complex recruiting RNase E near the complex. In this work, Lalaouna et al. claim that Hfq is dispensable for the duplex formation between certain sRNAs and target mRNAs in vivo by using pull-down assays of MS2-tagged sRNAs in combination with high-throughput RNA sequencing. They show that Hfq binding to target mRNAs is required for sRNA-induced mRNA degradation, proposing that a primary function of Hfq is to promote sRNA-induced cleavage of mRNA targets by RNase E.

Major points:

This work is a challenge to the well-established concept regarding the role of Hfq in sRNA-mediated gene regulation in bacteria. The central experiment is the pull-down assay of MS2-tagged RyhB. The authors demonstrate that MS2-tagged RyhB is able to pull-down its target mRNAs such as sodB mRNA in lysate prepared from cells expressing MS2-tagged RyhB and argue that RyhB can base-pair with sodB mRNA even in the Δhfq background (Fig. 1). The data suggest but do not prove that sRNAs can base pair with cognate target mRNAs in the absence of Hfq. It is dangerous to conclude that Hfq is not essential for the duplex formation in vivo at this stage because there are possible alternative interpretations of the results. For example, MS-tagged RyhB could co-purify sodB mRNA somehow in the base-pairing independent manner. To test this possibility, the pull-down assay using mutated MS-tagged RyhB which lacks the base-pairing ability is at least needed. Second, it is possible that MS-tagged RyhB could form RNA duplex with sodB mRNA in the absence of Hfq during the experiment and this may not faithfully reflect the real event in cells. Indeed, it is well known that sRNAs could base pair with cognate mRNAs without Hfq in vitro under some conditions such as a longer incubation or a higher temperature. The authors do not exclude this simple possibility.

Other points:

  • The data shown in Fig. 2 obtained by the super-resolution imaging approach are not clear to argue the significant colocalization of RyhB and sodB mRNA compared to the control.
  • The effect of RyhB expression on the SodB-LacZ translational fusion is significantly reduced in Δhfq background (Fig. 3B). Considering that RyhB expression also reduces sodB-lacZ transcriptional fusion even in Δhfq background (Fig. 3A), the repressive effect of RyhB expression on sodB translation is apparently less significant. Thus, it is not convincing to conclude from the data shown in Fig. 3 that RyhB blocks translation of sodB mRNA in the absence of Hfq. The data are rather consistent with the view that the sodB translation is inhibited by RyhB more efficiently in the presence of Hfq.
  • The results shown in Fig. 5 suggest that the Y25D mutation affect both mRNA degradation and translational block.

The data are overall ambiguous and not sufficient enough to support the important conclusion that RyhB forms RNA duplex with target mRNAs and inhibits their translation in the absence of Hfq in vivo to exert their regulatory function. The authors should be more careful to interpret the results.

Author Response

Major points:

This work is a challenge to the well-established concept regarding the role of Hfq in sRNA-mediated gene regulation in bacteria. The central experiment is the pull-down assay of MS2-tagged RyhB. The authors demonstrate that MS2-tagged RyhB is able to pull-down its target mRNAs such as sodB mRNA in lysate prepared from cells expressing MS2-tagged RyhB and argue that RyhB can base-pair with sodB mRNA even in the Δhfq background (Fig. 1). The data suggest but do not prove that sRNAs can base pair with cognate target mRNAs in the absence of Hfq. It is dangerous to conclude that Hfq is not essential for the duplex formation in vivo at this stage because there are possible alternative interpretations of the results.

Authors answer: We agree with the reviewer and certainly do not want to overinterpret our data especially in the first figure of the manuscript. This is why we used cautious language like “Hfq might not be required” or “Hfq might not be essential for pairing” and “results presented above suggested RyhB could bind to both sodB and sdhC target mRNAs without Hfq” throughout the manuscript but especially when describing the results of Figure 1.

For example, MS-tagged RyhB could co-purify sodB mRNA somehow in the base-pairing independent manner. To test this possibility, the pull-down assay using mutated MS-tagged RyhB which lacks the base-pairing ability is at least needed.

Authors answer: We understand the point of the reviewer that this is an excellent experiment to perform. However, we believe that presenting such data in a forthcoming manuscript will be more appropriate.

Second, it is possible that MS-tagged RyhB could form RNA duplex with sodB mRNA in the absence of Hfq during the experiment and this may not faithfully reflect the real event in cells. Indeed, it is well known that sRNAs could base pair with cognate mRNAs without Hfq in vitro under some conditions such as a longer incubation or a higher temperature. The authors do not exclude this simple possibility.

Authors answer: We thank the reviewer for this comment because too many papers using in vitro experiments is the motivation behind this manuscript. This is why we used in vivo single-molecule approach to test the role of Hfq in sRNA binding a target. Furthermore, we designed in vivo experiments showing that the sRNA is still able to repress translation in vivo in the absence of Hfq. We also made sure that the expression of the sRNA RyhB in vivo is close to the endogenous level such as our results in Figure 3 (both lanes 5). Thus, in contrast to the reviewer’s claim, we do believe that our multiple in vivo experiments have excluded the possibility of sRNA forming a duplex with a target mRNA due to in vitro conditions.  

Other points:

The data shown in Fig. 2 obtained by the super-resolution imaging approach are not clear to argue the significant colocalization of RyhB and sodB mRNA compared to the control.

Authors answer: The results presented in Figure 2 are indeed subtle to distinguish differences. This is why we used densitometry of the signals to represent the data in Figure 2C, which convincingly show an expected higher rate of colocalization between RyhB and sodB130-lacZ compared to RyhB and ptsG.  

The effect of RyhB expression on the SodB-LacZ translational fusion is significantly reduced in Δhfq background (Fig. 3B). Considering that RyhB expression also reduces sodB-lacZ transcriptional fusion even in Δhfq background (Fig. 3A), the repressive effect of RyhB expression on sodB translation is apparently less significant. Thus, it is not convincing to conclude from the data shown in Fig. 3 that RyhB blocks translation of sodB mRNA in the absence of Hfq. The data are rather consistent with the view that the sodB translation is inhibited by RyhB more efficiently in the presence of Hfq.

Authors answer: The challenge in a successful experiment using hfq mutants resides in a careful observation of the level of expressed sRNA. Because Hfq stabilizes many sRNAs, including RyhB, its absence makes those same sRNAs quite unstable. Thus, we took great care to express RyhB at a native level despite the absence of Hfq. Concerning the reviewer comment about the less significant repressive effect of RyhB on sodB translation, we respectfully disagree. Figure 3 clearly shows that sodB translation is repressed to 50% of its WT level in absence of Hfq. Because the transcriptional sodB-lacZ is repressed by 25% in the same conditions, we can only conclude that translation is directly repressed by RyhB: “Results in Figure 3B indicate a significant reduction of the SodB430-LacZ in the Δhfq cells in presence of RyhB (50%). The stronger repression observed in WT cells (70%), could be due to higher RyhB level in WT versus Δhfq cells (compare lanes 2 and 4, Northern blot) or a more active degradation of the target (Figure 3A)”.

The results shown in Fig. 5 suggest that the Y25D mutation affect both mRNA degradation and translational block.

Authors answer: We thank the reviewer for his supportive comment and appreciation of our results.

The data are overall ambiguous and not sufficient enough to support the important conclusion that RyhB forms RNA duplex with target mRNAs and inhibits their translation in the absence of Hfq in vivo to exert their regulatory function. The authors should be more careful to interpret the results.

Authors answer: We appreciate the reviewer’s comment. We have downplayed some statement in the Discussion by modifying the following sentence: “Contrary to this, we provide evidence from multiple approaches that cellular Hfq is probably dispensable for RyhB, RybB, and DsrA sRNAs binding to many of their target mRNAs in vivo”.

Reviewer 2 Report

Summary:

The manuscript investigates the role that the RNA chaperone Hfq plays in sRNA-mediated regulation of gene expression in Escherichia coli. Specifically, the authors provide evidence that the primary function of Hfq may not be the facilitation of sRNA–mRNA pairing, as is often assumed. The manuscript focuses on the well-studied RyhB–sodB and RyhB–sdhC sRNA–mRNA interactions and reports that Hfq is not required for RyhB and sodB/sdhC to interact in vivo, rather, Hfq is required for RNase E-mediated degradation of the sodB/sdhC mRNA. Furthermore, Hfq is required to bind to sodB/sdhC mRNA to mediate this effect. Finally, the manuscript reports additional sRNA–mRNA interactions that can be detected in vivo in the absence of Hfq, again suggesting that mediating the base-pairing interaction is not the critical function of Hfq in these cases.

Broad comments:

The manuscript is well-written and logical. Some of the findings and conclusions are not particularly surprising given what is already known about RyhB and its mechanism-of-action. However, the use of technological advances such as MS2-tagged RNAs and super-resolution microscopy does further the understanding of the role of Hfq in RyhB-mediated gene expression regulation.  

Given the focus on RyhB, I would have liked a more thorough introduction to RyhB to have been included in the Introduction. In particular, it would have been helpful to clearly state what is, and isn’t, known about the interplay between Hfq, RyhB and RyhB targets. This would help to highlight the novelty of the work presented.

It would be helpful to be explicit about the strain background that is used in each experiment and explain the choice of strain background. I found it difficult to keep track of whether Hfq, RNase E and/or the sRNA were present or absent in each experiment. For example, in Figure 1A, I believe that the ryhB and rne genes are present. RyhB presumably wouldn’t be produced under the growth conditions used, but it would be useful to make this clear, and it would be useful to discuss what RNase E would be expected to do in the experiment.   

It would also be helpful to provide a brief introduction to the different sRNAs/mRNAs as they are introduced. For example, clearly stating that sodB and sdhC mRNAs are known targets of RyhB and that ptsG mRNA is a target of SgrS.

Specific comments: 

Line 179 – Please explain why the sodB130-lacZ fusion is RNase E-resistant (expected cleavage site not present) and/or provide a reference.

Figure 2 – It might be helpful to also indicate the binding sites of the sodB130-lacZ and ptsG probes.

Lines 273-275/Supplementary Figure S4 – The EMSA for sodB130MH does not look like it has plateaued. Therefore, you cannot accurately calculate a Kd. As the point is to provide a comparison, perhaps it would be better to indicate that the Kd is at least 314 nM for the mutant.

Figure 5 legend – Please provide a legend for part D (I presume it is quantitation of part C).

Lines 619-629 – Please check the super-resolution imaging methods. I couldn’t find details about when/where the probes were added, or their concentration. Please include this information.

Figure S1 legend – Please correct the legend. Part B is not quite the same as part A with MS2-SgrS as bait because the probed mRNA is ptsG rather than sodB and sdhC etc.

Author Response

Broad comments:

The manuscript is well-written and logical. Some of the findings and conclusions are not particularly surprising given what is already known about RyhB and its mechanism-of-action. However, the use of technological advances such as MS2-tagged RNAs and super-resolution microscopy does further the understanding of the role of Hfq in RyhB-mediated gene expression regulation. 

Authors answer: We thank the reviewer for the kind and supportive words.

Given the focus on RyhB, I would have liked a more thorough introduction to RyhB to have been included in the Introduction. In particular, it would have been helpful to clearly state what is, and isn’t, known about the interplay between Hfq, RyhB and RyhB targets. This would help to highlight the novelty of the work presented.

Authors answer: Excellent suggestion. However, we feel there is enough information in the introduction to appreciate the results presented here.

It would be helpful to be explicit about the strain background that is used in each experiment and explain the choice of strain background. I found it difficult to keep track of whether Hfq, RNase E and/or the sRNA were present or absent in each experiment. For example, in Figure 1A, I believe that the ryhB and rne genes are present. RyhB presumably wouldn’t be produced under the growth conditions used, but it would be useful to make this clear, and it would be useful to discuss what RNase E would be expected to do in the experiment.  

Authors answer: We understand the reviewer’s concern. However, we find that adding the names of the strains would make the reading of the manuscript more difficult. Thus, we opted for using a leaner approach and clearly indicate the relevant phenotype used in each figure of the manuscript.

It would also be helpful to provide a brief introduction to the different sRNAs/mRNAs as they are introduced. For example, clearly stating that sodB and sdhC mRNAs are known targets of RyhB and that ptsG mRNA is a target of SgrS.

Authors answer: We added the following text to make it clearer which sRNAs regulated whichtarget mRNAs. “With more than 25 known target mRNAs, the iron-responsive sRNA RyhB, has one of the largest regulons in E. coli, including sodB and sdhC transcripts” and RyhB and the negative control ptsG mRNA, which not regulated by RyhB,  were imaged together as negative control to account for random colocalization”.

Specific comments:

Line 179 – Please explain why the sodB130-lacZ fusion is RNase E-resistant (expected cleavage site not present) and/or provide a reference.

Authors answer: We added a reference that described why the sodB130-lacZ fusion is resistant to RNase E.

Lines 273-275/Supplementary Figure S4 – The EMSA for sodB130MH does not look like it has plateaued. Therefore, you cannot accurately calculate a Kd. As the point is to provide a comparison, perhaps it would be better to indicate that the Kd is at least 314 nM for the mutant.

Authors answer:

Figure 5 legend – Please provide a legend for part D (I presume it is quantitation of part C).

Authors answer: We added: “(D) Quantification of the RyhB RNA level normalized to the 5S rRNA level shown in (C). Error bars represent the standard deviation of three biological replicates”.

Lines 619-629 – Please check the super-resolution imaging methods. I couldn’t find details about when/where the probes were added, or their concentration. Please include this information.

Authors answer: We modified the following sentence in section 5.7 to include the information: “60 µL of these permeabilized cells were washed with FISH wash solution (10% formamide in 2X SSC) and resuspended in 15 µL of hybridization buffer (10% dextran sulfate, 10% formamide in 2X SSC) containing the specific probes”.

Figure S1 legend – Please correct the legend. Part B is not quite the same as part A with MS2-SgrS as bait because the probed mRNA is ptsG rather than sodB and sdhC etc.

Authors answer: We corrected the legend.

Reviewer 3 Report

Summary

Although Hfq is one of the most widely studied bacterial RNA-binding proteins we still do not fully understand its functions. It is largely known that Hfq is a main factor in sRNA biology and sRNA-mediated pathways and is commonly believed that Hfq is required for the interaction between a sRNA and its target mRNA(s). The work by Lalaouna and colleagues now adds important data on this view, demonstrating that Hfq can be dispensable for the formation of certain sRNA:mRNA pairs in vivo.

This is a very well structured and nicely written manuscript, which I really enjoyed reading. Through the use of an elegant set of experiences performed in vivo, the authors make a strong case for their claiming that sRNA:mRNA interactions can occur in the absence of Hfq. Having said that, I believe that there are some points that could still be clarified and/or addressed in discussion.

Major points

  1. The MAPS method previously developed by the authors, that employs MS2-sRNA affinity purification coupled with RNA sequencing, has proven to be a fantastic tool for the identification of new target mRNAs for numerous sRNAs. This is largely validated. Nevertheless, I have some doubts if this same approach can be used to rule out a role for Hfq in promoting sRNA:mRNA base pair in vivo. The reason for my concern is that this method uses induction from a strong promoter for sRNA expression, resulting in very high levels of a given sRNA.

Couldn´t the high increase in sRNA concentration simply overcome the need of an RNA chaperone to promote the base pairing with a target mRNA, by strongly increasing the likelihood of interaction between the two RNA species? In other words, how physiologic is your system? Can we really exclude Hfq from participating in sRNA:mRNA interaction? For example, the induction of MS2-RyhB in your experimental setting is much stronger than the induction of RyhB when iron is removed from the medium (which seems a more physiological condition). I think the authors should address this in discussion, either to offer this alternative view or, even better, to rule out this hypothesis.

  1. According to the authors, the major role of Hfq would be to promote the sRNA-mediated degradation of the target mRNA by RNase E. However, can the authors really generalize this? This is well known to happen with RyhB but is it known for all other sRNAs (namely the ones here presented, RybB and DsrA)? In this case, please add references for this or, alternatively, you would need to show that RNase E is implicated in the sRNA-dependent degradation of other mRNAs. My main point is that you should not merely focus on RNase E-mediated degradation. I do not see why RNase III (or other endos) would be excluded from this degradation (see for example Reuscher and Klug, 2021).

  1. A point that I also do not think is sufficiently addressed is the fact that Hfq is required for sRNA stability, as Hfq protects sRNAs from nucleolytic attack. This is very important in all your work, as in the del-hfq background you can clearly detect reduced levels of sRNAs compared to the wild-type. Most of the times you report that Hfq is important for stabilization of the sRNAs but you can add more information in here. You should, for example, include in the manuscript that PNPase is a major nuclease involved in the degradation of sRNAs, namely RyhB as described in Andrade et al 2012 and more recently in Chen et al, 2021. And I imagine that other enzymes may participate in this degradation as well (Quendera et al, 2020). I think this should be addressed at least in introduction (85-86) and conclusion (lines 509-510) sections.

  1. I was a bit confused which are the WT and del-hfq backgrounds used throughout the work, and I would ask for the name of the strains provided in the legends and/or when describing the results. This is easier than searching the Table S1 for the correct strain.

  1. In Figure 3A, inactivation of Hfq results in higher levels of sodB mRNA when compared to the wild type strain when RyhB is induced. However, it is clear that sodB mRNA is still degraded in the absence of Hfq. Does this imply that RNase E-mediated degradation of sodB mRNA do not necessarily depend on Hfq? Or that other factors that not RNase E are involved in this degradation? This needs to be addressed, as it questions how essential the Hfq/RNase E cooperation for sRNA-induced mRNA degradation is. Hence, I think the title of section 2.4 “Efficient RyhB-induced degradation of sodB target mRNA requires Hfq” is too strong as it is. I would suggest changing “require Hfq” to “is facilitated by Hfq”.

  1. The authors clearly showed that mutation of the Hfq-binding site in the sodB mRNA do not affect RyhB-induced translation block (Figure 4). This implies that this mutation does not significantly alter the secondary structure of the sodB mRNA around the start codon, thus not affecting RyhB binding. I do not think this is well stressed in the manuscript and perhaps this is useful. In addition, RNA mapping from Figure S4 could not be used to backup this idea?

  1. The indication that RyhB levels increase in Y25D Hfq mutant strain is very interesting. Do the authors know if this is a more general effect on other sRNAs or if this is specific to RyhB?

  1. In Figure 6D, the MS2-DsrA pulldown shows reduced levels of hns mRNA recovered from the del-hfq background compared to the wild-type strain, in contrast to what is shown to MS2-RyhB and MS2-RybB target mRNAs. Why is this?

Minor points

Line 172: shiA is not presented in the Figure 1C or 1D

Line 303: change “fq” to “Hfq”

Figure 3: Add symbols (-) and (+) in the legend

Figure 5C: Add the half-lives in the legend or in results

Figure S1: probably needs to remove “with the previously known target mRNA (ompC)

Author Response

Major points

The MAPS method previously developed by the authors, that employs MS2-sRNA affinity purification coupled with RNA sequencing, has proven to be a fantastic tool for the identification of new target mRNAs for numerous sRNAs. This is largely validated. Nevertheless, I have some doubts if this same approach can be used to rule out a role for Hfq in promoting sRNA:mRNA base pair in vivo. The reason for my concern is that this method uses induction from a strong promoter for sRNA expression, resulting in very high levels of a given sRNA.

Authors answer: The challenge in a successful experiment using hfq mutants resides in a careful observation of the level of expressed sRNA. Because Hfq stabilizes many sRNAs, including RyhB, its absence makes those same sRNAs quite unstable. Thus, we took great care to express RyhB at a native level despite the absence of Hfq. Figure 3 demonstrates the levels of induced RyhB were comparable to endogenous RyhB levels. As such, the plasmid-expressed RyhB levels were equivalent to endogenous levels.

Couldn´t the high increase in sRNA concentration simply overcome the need of an RNA chaperone to promote the base pairing with a target mRNA, by strongly increasing the likelihood of interaction between the two RNA species? In other words, how physiologic is your system? Can we really exclude Hfq from participating in sRNA:mRNA interaction? For example, the induction of MS2-RyhB in your experimental setting is much stronger than the induction of RyhB when iron is removed from the medium (which seems a more physiological condition). I think the authors should address this in discussion, either to offer this alternative view or, even better, to rule out this hypothesis.

Authors answer: We agree with the reviewer, and this is why we do not use RyhB-MS2 for all our experiments. Indeed, we used endogenously-expressed RyhB in many of our experiments. For example, Figure 2 shows an endogenously-expressed RyhB able to bind a sodB-lacZ construct in a hfq mutant. This previous experiment was performed in single-molecule resolution using super-resolution microscopy. Moreover, Figure 3 shows endogenous levels of RyhB repressing translation from sodB-LacZ translational construct in a hfq mutant.

We also addressed this point several times in the text. In paragraph 2.7: “Importantly, to demonstrate that these results were not dependent on the overproduction of RyhB from a heterologous (pBAD) promoter, we performed a similar experiment but with RyhB expressed from the endogenous locus during iron starvation induced by the iron chelator 2,2’-dipyridyl (DIP; Figures S5D and S5E). In agreement with the results shown above, endogenous levels of RyhB produced during iron starvation efficiently repress both SodB430MH-LacZ and SdhC258MH-LacZ translation. These results suggest that abrogating Hfq binding to these target mRNAs does not interfere with RyhB base pairing”.

In the Discussion: “Finally, using the recently developed super-resolution imaging method (Fei et al., 2015), that allows us to image and quantify sRNA:mRNA complex formation, we confirmed that endogenously-expressed RyhB (induced by iron starvation) and target mRNA sodB can form a complex in the presence or absence of Hfq in vivo (Figure 2)”.

Also in the Discussion: “Importantly, we demonstrated in this study that endogenously-expressed RyhB can base pair with and regulate target mRNAs in the absence of Hfq or when Hfq binding sites on the target were mutated”.

According to the authors, the major role of Hfq would be to promote the sRNA-mediated degradation of the target mRNA by RNase E. However, can the authors really generalize this? This is well known to happen with RyhB but is it known for all other sRNAs (namely the ones here presented, RybB and DsrA)? In this case, please add references for this or, alternatively, you would need to show that RNase E is implicated in the sRNA-dependent degradation of other mRNAs. My main point is that you should not merely focus on RNase E-mediated degradation. I do not see why RNase III (or other endos) would be excluded from this degradation (see for example Reuscher and Klug, 2021).

Authors answer: Throughout the manuscript, we have made it clear that the observed sRNA-mediated degradation results remain specific to RyhB:

In the title: “Binding of the RNA chaperone Hfq on target mRNAs promotes the small RNA RyhB-induced degradation in Escherichia coli”.

In par 2.4: “These results are consistent with the idea that Hfq is required for efficient RyhB-induced degradation of sodB target mRNA”.

In par 2.6: “These data suggest that the Hfq binding site located upstream of the RBS is critical for RyhB-induced degradation of both sodB and sdhC target mRNAs”.

In par 2.8: “We also observed some remaining endogenous sodB RNA using Northern blot analysis (compare lanes 1 and 3), suggesting a reduced RyhB-induced sodB degradation”.

We have changed a sentence in the Discussion to make our statement more RyhB-specific: “Instead, loss of Hfq binding on target mRNAs (Figure 4D and 5A) reduced RyhB-dependent mRNA degradation”.

However, we do not argue that sRNA-induced degradation of target mRNA is also affected in RybB and DsrA cases. We rather describe how these sRNA can bind their cognate targets in absence of wild-type Hfq as RyhB.

A point that I also do not think is sufficiently addressed is the fact that Hfq is required for sRNA stability, as Hfq protects sRNAs from nucleolytic attack. This is very important in all your work, as in the del-hfq background you can clearly detect reduced levels of sRNAs compared to the wild-type. Most of the times you report that Hfq is important for stabilization of the sRNAs but you can add more information in here. You should, for example, include in the manuscript that PNPase is a major nuclease involved in the degradation of sRNAs, namely RyhB as described in Andrade et al 2012 and more recently in Chen et al, 2021. And I imagine that other enzymes may participate in this degradation as well (Quendera et al, 2020). I think this should be addressed at least in introduction (85-86) and conclusion (lines 509-510) sections.

Authors answer: This is an interesting point to make but as an editorial choice, we have decided against this because we do not want to dilute the message of the manuscript. However, we will address this in a forthcoming article mentioned below in the comment about RyhB stability.

I was a bit confused which are the WT and del-hfq backgrounds used throughout the work, and I would ask for the name of the strains provided in the legends and/or when describing the results. This is easier than searching the Table S1 for the correct strain.

Authors answer: We understand the reviewer’s concern. However, we find that adding the names of the strains would make the reading of the manuscript more difficult. Thus, we opted for using a leaner approach and clearly indicate the relevant phenotype used in each figure of the manuscript.

In Figure 3A, inactivation of Hfq results in higher levels of sodB mRNA when compared to the wild type strain when RyhB is induced. However, it is clear that sodB mRNA is still degraded in the absence of Hfq. Does this imply that RNase E-mediated degradation of sodB mRNA do not necessarily depend on Hfq? Or that other factors that not RNase E are involved in this degradation? This needs to be addressed, as it questions how essential the Hfq/RNase E cooperation for sRNA-induced mRNA degradation is. Hence, I think the title of section 2.4 “Efficient RyhB-induced degradation of sodB target mRNA requires Hfq” is too strong as it is. I would suggest changing “require Hfq” to “is facilitated by Hfq”.

Authors answer: We have changed the sentence to : “Efficient RyhB-induced degradation of sodB target mRNA is promoted by Hfq”.

The authors clearly showed that mutation of the Hfq-binding site in the sodB mRNA do not affect RyhB-induced translation block (Figure 4). This implies that this mutation does not significantly alter the secondary structure of the sodB mRNA around the start codon, thus not affecting RyhB binding. I do not think this is well stressed in the manuscript and perhaps this is useful. In addition, RNA mapping from Figure S4 could not be used to backup this idea?

Authors answer: We appreciate the reviewer’s suggestions and added a sentence to the text: “Moreover, our results suggest that secondary structures in the translation start region of sodBMH are not significantly altered by the mutation (Figure S4C)”.

The indication that RyhB levels increase in Y25D Hfq mutant strain is very interesting. Do the authors know if this is a more general effect on other sRNAs or if this is specific to RyhB?

Authors answer: We indeed observed this and wrote this paragraph in section 2.8: “We next investigated in vivo RyhB stability in the presence of Hfq Y25D mutant. Because RyhB was previously demonstrated to co-degrade with target mRNAs (Masse et al., 2003), we reasoned that absence of Hfq binding target mRNAs (hfqY25D background) might reduce degradation rate of RyhB. As shown in Figure 5C, the turnover of RyhB sRNA decreases significantly in the presence of Hfq Y25D as compared to WT background. This indicates that reducing degradation of target mRNAs in a hfqY25D background decreases turnover of RyhB”. We plan to investigate other sRNAs in a forthcoming manuscript.

In Figure 6D, the MS2-DsrA pulldown shows reduced levels of hns mRNA recovered from the del-hfq background compared to the wild-type strain, in contrast to what is shown to MS2-RyhB and MS2-RybB target mRNAs. Why is this?

Authors answer: The MAPS technique has no crosslinking step and as such does not guarantee that a sRNA-mRNA complex remains stable during following steps of extraction. We also expect that target mRNAs length, structure, ribosome coverage might play a role in extraction procedure. As each target mRNA might react in a specific fashion, it is difficult to predict the yield for each extracted RNAs. It is also possible that Hfq-driven RNase E recruitment plays a role in making some target much more stable in the absence of Hfq. DsrA is also known to cleave and produce a truncated and smaller version, which can reduce the binding of target in the process.

Minor points

Line 172: shiA is not presented in the Figure 1C or 1D

Authors answer: We removed it from the list.

Line 303: change “fq” to “Hfq”

Authors answer: Done

Figure 3: Add symbols (-) and (+) in the legend

Authors answer: Done

Figure 5C: Add the half-lives in the legend or in results

Authors answer: Done

Figure S1: probably needs to remove “with the previously known target mRNA (ompC)

Authors answer: Done